# Unsupervised object-centric video generation and decomposition in 3D

**Paul Henderson**
IST Austria
`paul@pmh47.net`

**Christoph H. Lampert**
IST Austria
`chl@ist.ac.at`

## Abstract

A natural approach to generative modeling of videos is to represent them as a composition of moving objects. Recent works model a set of 2D sprites over a slowly-varying background, but without considering the underlying 3D scene that gives rise to them. We instead propose to model a video as the view seen while moving through a scene with multiple 3D objects and a 3D background. Our model is trained from monocular videos without any supervision, yet learns to generate coherent 3D scenes containing several moving objects. We conduct detailed experiments on two datasets, going beyond the visual complexity supported by state-of-the-art generative approaches. We evaluate our method on depth-prediction and 3D object detection—tasks which cannot be addressed by those earlier works—and show it out-performs them even on 2D instance segmentation and tracking.

## 1 Introduction

In recent years, there has been considerable interest in *object-centric* generative models of images and videos—that is, generative models whose latent structure explicitly represents their composition from multiple objects or regions (e.g. [12, 13, 15, 24, 29]). These methods allow segmentation of input videos or images into objects, and generation of new ones. This is a natural structure to adopt, as it mirrors the way humans understand the world [33], as well as capturing important aspects of the causal process by which images are formed. Existing approaches treat objects either as components of 2D spatial mixtures, or as 2.5D stacks of sprites, which reflects how they appear when imaged by a camera. Crucially however, it does *not* reflect the underlying physical structure of the world, which of course consists of solid 3D objects situated in 3D space.

In this work, we take the natural next step—we develop an object-centric generative model over videos that explicitly models the 3D scenes they show. Our model is trained purely from unannotated monocular videos, without any 3D data. Nonetheless, it learns to decompose videos into multiple 3D foreground objects and a 3D background, and learns to generate videos showing coherent scenes.

To achieve this, we design a novel generative model that represents videos as the view from a camera moving through a 3D scene composed of multiple objects and a background (Sec. 2). It has a single latent embedding that learns to capture the space of allowable scene structures, which is important to avoid sampling implausible layouts such as intersecting objects [12]. This embedding is interpreted by a structured decoder that processes objects independently and compositionally. Each object is endowed with an appearance embedding, and a temporally-varying 3D location and pose. After decoding their appearance embeddings to 3D shapes and textures, the objects and background are differentiably rendered to give the final frames. This lets us train the model like a variational autoencoder (VAE) [28], to reconstruct its training videos in terms of their latent embeddings (Sec. 3).

By decoding objects independently, and disentangling pose from appearance, we introduce inductive biases that allow a distribution of videos to be represented efficiently and compositionally [45, 46], without needing to separately model all possible combinations of objects, shapes, textures, and poses.

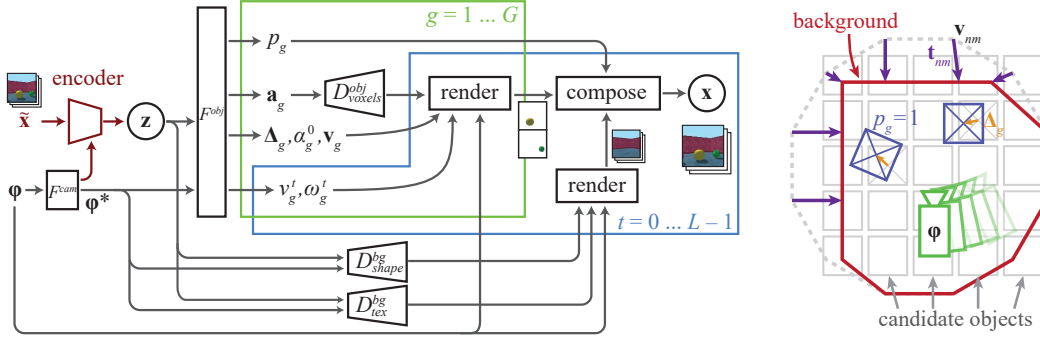

Figure 1: **Left:** Our generative model of videos has a single latent variable $\mathbf{z}$, which is decoded by $F^{\mathrm{obj}}$ to parameters of $G$ objects. The contents of the green plate are replicated once per object; the contents of the blue plate are replicated once per frame. Each object has pose parameters and an appearance embedding $\mathbf{a}_g$, which is itself decoded (by $D^{\mathrm{obj}}_{\mathrm{voxels}}$) to an explicit voxel representation of its 3D shape and color. $D^{\mathrm{bg}}_{\mathrm{shape}}$ and $D^{\mathrm{bg}}_{\mathrm{tex}}$ map $\mathbf{z}$ to the shape and texture of a 3D background. The objects are rendered individually based on their pose and the camera parameters $\phi$, then composed over the background to give the final frames $\mathbf{x}$. We add an encoder model (red) that predicts the latent variable corresponding to a given video $\tilde{\mathbf{x}}$. **Right:** Plan view of our 3D scene structure, viewed by a camera (green), which moves over time according to parameters $\phi$. We define a grid of $G$ candidate objects in 3D space (gray boxes), and a spherical shell for the background (gray dashed). Presence indicators $p_g$ specify whether the object in each cell is present; here the blue ones are. Each is displaced by $\Delta_g$ from the center of its cell (orange), and rotated by $\alpha_g$; the position and rotation then vary over time. The background is deformed (purple arrows) into the correct shape (red) to capture the surrounding environment

Moreover, by treating the objects and background as 3D, we automatically capture the subspace of appearance variation that arises due to viewpoint changes, which would otherwise need to be explicitly learnt by the model and encoded in its latent space. This is important as it eases the necessary trade-off between capacity and expressiveness—a generic model powerful enough to explain appearance variation due to viewpoint changes may more easily learn to model several objects together as one (an undesirable outcome). Factoring out this aspect of variability allows our approach to handle videos with significantly higher visual complexity than previous methods [9, 24].

We conduct extensive experiments (Sec. 5) on two datasets of videos with both stationary and moving objects. We show that our method can:

- decompose a given video into its constituent objects and background, by predicting segmentation masks and tracking objects over time;
- determine its 3D structure, by predicting depth and 3D bounding boxes; and
- generate coherent videos showing objects moving in 3D space over a 3D background.

In short, our contribution is *the first object-centric generative model of videos that explicitly reasons in 3D space, learning in an unsupervised fashion to decompose scenes into 3D objects, and to generate new, plausible samples*.

## 2 Generative model

Our generative process (Fig. 1) for a video $\mathbf{x} \in \mathbb{R}^{L \times H \times W}$, of length $L$ frames and size $W \times H$ pixels, begins by sampling a $d$-dimensional Gaussian latent variable $\mathbf{z} \sim \mathcal{N}(\mathbf{0}, \mathbf{1})$. This embeds the full content of the scene shown in the video, and is decoded to yield separate embeddings and poses for the different objects, and the shape and texture of the background. The object embeddings are decoded independently to 3D shapes, represented as voxels or meshes. Then, the objects and background are rendered into the final video frames, and Gaussian pixel noise is added so the likelihood is well-defined. The camera is assumed to be at the origin at frame zero, hence the model is *ego-centric*—it represents each scene in a frame of reference centered on the viewer. The camera parameters $\phi$ are treated as a known conditioning variable, a natural assumption in many applications such as robotics. We flatten the camera matrices to a single vector and encode this with a fully-connected network

$F^{\mathrm{cam}}$ to give an embedding $\phi^* \in \mathbb{R}^c$. In the following subsections, we describe each component of the decoder in detail.[1]

## 2.1 Background

We model the background as a large shell around the initial location of the camera, represented by a triangle mesh. The decoder deforms and colors this mesh to represent complex environments. We choose a mesh because it is an efficient representation for large, continuous surfaces (in contrast to point-clouds which have holes, and voxels which have high memory requirements), and can faithfully represent environments both indoors (e.g. rooms) and outdoors (e.g. cityscapes). Specifically, a 2D transpose-convolutional network $D_{\mathrm{shape}}^{\mathrm{bg}} : \mathbb{R}^d \times \mathbb{R}^c \to \mathbb{R}^{N \times M \times 4}$ maps $\mathbf{z}$ and $\phi^*$ to a transform for each vertex, similar to works on single-image 3D reconstruction [26, 47]. Here, $N$ and $M$ index the vertices of a UV-sphere, i.e. a mesh of rectangular grid topology wrapped around and distorted to approximate a sphere. The transform for the $nm^{\mathrm{th}}$ vertex $\mathbf{v}_{nm}$ is a 4D vector $\mathbf{t}_{nm}$, specifying a radial scale factor and a 3D offset; the transformed vertex location is $\mathbf{v}_{nm} \exp(t_{nm}^0) + \gamma \tanh(\mathbf{t}_{nm}^{1...3})$, where $\gamma$ is a hyperparameter. This parameterization ensures that the geometry cannot become too irregular, but is still able to represent complex shapes. Another, similar decoder $D_{\mathrm{tex}}^{\mathrm{bg}}(\mathbf{z}, \phi^*)$ outputs an RGB texture that is rendered onto the mesh; this has $6\times$ higher spatial resolution than $D_{\mathrm{shape}}^{\mathrm{bg}}$, allowing it to capture finely-detailed appearance variations. We render the background by first transforming the vertices according to the camera parameters $\phi$ for each frame, and then applying a perspective projection to map them into image space. The resulting triangles are rasterized differentiably using the method of [19], and the texture is bilinearly sampled at every pixel to give video frames $\mathbf{x}_{\mathrm{bg}}$.

## 2.2 Objects

Inspired by [8, 24, 37], we define a grid of *candidate objects*—but unlike those works, this grid covers the 3D space of the scene, rather than 2D pixel space. Object positions are then specified as a displacement relative to the center of their respective grid cell. Binary variables indicate whether each candidate object is present in a given scene; in practice we relax these indicators to be continuous values in $[0, 1]$. This technique allows gradients to propagate through many possible object locations, avoiding cases where the model does not receive an informative gradient signal due mispredicting an object's location badly enough that there is no overlap between the reconstructed and original instance [24]. We let $G$ denote the number of cells in the grid, and $\mathbf{c}_g \in \mathbb{R}^3$ the center of the $g^{\mathrm{th}}$ cell.

A fully-connected decoder $F^{\mathrm{obj}} : \mathbb{R}^d \times \mathbb{R}^c \to \mathbb{R}^{G \times (e+8+2(L-1))}$ outputs for each object grid-cell $g$: a presence indicator $p_g \in [0, 1]$, an appearance embedding $\mathbf{a}_g \in \mathbb{R}^e$, an initial offset $\Delta_g \in \mathbb{R}^3$ from the grid-cell center, an initial velocity $\mathbf{v}_g \in \mathbb{R}^3$, an initial azimuth $\alpha_g^0 \in [0, 2\pi]$, and per-frame log-speeds $\nu_g^t \in \mathbb{R}$ and angular velocities $\omega_g^t \in \mathbb{R}$ for $0 < t < L$.[2] The time-varying azimuth of object $g$ for frames $t > 0$ is then given by $\alpha_g^t = \alpha_g^{t-1} + \omega_g^t$, and its location by $\mathbf{\Lambda}_g^t = \mathbf{c}_g + \Delta_g + \sum_{\tau=1}^t \left( \hat{\mathbf{v}} + \mathbf{v}_g \exp \nu_g^\tau \right)$, where $\hat{\mathbf{v}}$ is a bias hyperparameter.

The embeddings $\mathbf{a}_g$ are independently decoded to yield explicit representations of the 3D shapes of the corresponding objects. We now describe two different representations used in our experiments.

**Mesh objects.** Our first representation for 3D objects is a textured mesh. As for the background, we deform a UV-sphere, but with fewer vertices. We use two 2D transpose-convolutional decoder functions, $D_{\mathrm{shape}}^{\mathrm{obj}} : \mathbb{R}^e \to \mathbb{R}^{S \times T \times 4}$ and $D_{\mathrm{texture}}^{\mathrm{obj}} : \mathbb{R}^e \to \mathbb{R}^{3S \times 3T \times 3}$, for vertex transforms and RGB texture respectively. The 4D transform vectors are again interpreted as a radial scale and 3D offset. With this representation, the rendering process for an object is the same as for the background, after placing it according to $\mathbf{\Lambda}_g^t$ and $\alpha_g^t$. However, objects also have an associated presence variable $p_g$; when this is less than one, the object should be partially transparent, with other objects and the background visible through it. To achieve this, we first sort the objects according to how far their center point is from the camera, from farthest to nearest. Then, starting with the background image $\mathbf{x}_{\mathrm{bg}}$, we render each object in turn overlaying the previous ones, alpha blending it with weight $p_g$.

**Voxel objects.** Our second representation is a cuboidal grid of voxels, produced by a transpose-convolutional decoder $D_{\mathrm{voxels}}^{\mathrm{obj}} : \mathbb{R}^e \to \mathbb{R}^{V^3 \times 4}$, where $V$ is the voxel resolution, and the four output

channels represent RGB color and opacity. In order to efficiently render the $g^{\text{th}}$ object into the $t^{\text{th}}$ frame, we first find the region $R$ of the frame covered by its projected voxel cuboid. We also calculate its circumscribing ellipsoid $E$ in 3D space. Then, for each pixel in $R$, we cast a ray into the scene (by inverting the camera transform and perspective projection defined by $\phi_t$), and solve for its two intersections $\mathbf{i}_1$, $\mathbf{i}_2$ with $E$. Let $T_g^t$ be the homogeneous transform matrix corresponding to the object's pose $(\mathbf{\Lambda}_g^t, \alpha_g^t)$. We generate $K$ equally-spaced points $\mathbf{p}_k$ between $\mathbf{i}_1$ and $\mathbf{i}_2$, and calculate corresponding voxel coordinates $\mathbf{p}_k' = \frac{1}{s}(T_g^t)^{-1}\mathbf{p}_k + \frac{V}{2}$, where $s$ is the spacing of the voxels. We then trilinearly sample the voxel color and opacity values at each location $\mathbf{p}_k'$. We found $K = \lfloor \frac{4}{3}V \rfloor$ gives a satisfactory trade-off between quality and performance. Iterating from farthest-to-nearest, we then alpha-blend these samples over the background. Similar to meshes, we also perform an outer iteration over objects from farthest-to-nearest, and compose the results.

## 3 Training and inference

Our model is trained from a dataset of videos $\{\tilde{\mathbf{x}}\}$, without supervision. Directly maximizing the likelihood of the training data is intractable due to the latent variable $\mathbf{z}$; hence, we use stochastic gradient variational Bayes [28, 38]. Specifically, we maximize the evidence lower bound (ELBO), a variational bound on the log-likelihood of the training data, where the true posterior $P(\mathbf{z}|\tilde{\mathbf{x}})$ is replaced by a variational approximation. This variational posterior is parameterized by an encoder network, that predicts what distribution on $\mathbf{z}$ will reconstruct each input video $\tilde{\mathbf{x}}$. Our encoder network is a 3D (spatiotemporal) CNN, with alternating spatial and temporal convolutions, ReLU activations, and group-normalization; $\phi^*$ is given as an extra input to the first fully-connected layer. The posterior is assumed to be a diagonal Gaussian, so the encoder outputs a $d$-dimensional mean and standard deviation. Note that this architecture is much simpler than those used in similar works [24, 29], which have a recurrent attentive encoder. Much of their complexity stems from how they model objects appearing and disappearing; we instead assume that objects persist through time, but are allowed to move in and out of the camera view. Following [22], we weight the KL-divergence term in the ELBO according to a hyperparameter $\beta$; following [19], we augment the Gaussian pixel likelihood with a scale-space pyramid, to help avoid local optima. We use Adam [27] for optimization.

**Regularization.** Inferring 3D structure from 2D videos is inherently ambiguous, so regularization is important to avoid degenerate solutions. We therefore add the following loss terms:

- L1 regularization on the magnitude of the object velocities: $\frac{1}{LG}\sum_{t,g}\|\mathbf{v}_g\nu_g^t\|$. This discourages local minima where the model fails to track objects.
- hinge regularization on the presence variables for mesh objects: $\frac{1}{G}\sum_g \max\{0,\, 0.3 - p_g\}$. This discourages them from disappearing early in optimization, before their shape has adapted.
- inspired by classical works on Markov random fields for image segmentation [5, 39], we penalize edges in the reconstructed foreground mask for occurring in areas of the original image that have small gradients. This discourages undesirable but otherwise-valid solutions where an object is in front of an untextured surface, and parts of that surface are incorporated in the object rather than the background. Specifically, let $K_G$ be a $5 \times 5$ Gaussian smoothing kernel, and $D_x, D_y$ be central-difference derivative kernels. Let $\mathbf{m}$ be the predicted foreground mask for an input video $\tilde{\mathbf{x}}$; this is given by rendering all the $G$ predicted objects (like when reconstructing $\tilde{\mathbf{x}}$), but over a black background and with their color set to white. We then minimize

$$\frac{1}{L}\sum_t \int_\Omega \left\{ \left|D_x * \mathbf{m}^t\right| \exp\!\left(-\zeta \left|D_x * K_G * \mathbf{x}^t\right|\right) + \left|D_y * \mathbf{m}^t\right| \exp\!\left(-\zeta \left|D_y * K_G * \mathbf{x}^t\right|\right) \right\} d\mathbf{p} \quad (1)$$

where $t$ indexes frames, $\mathbf{p} \in \Omega$ ranges over pixels in a frame and $\zeta$ is a hyperparameter.
- standard mesh regularizers [25, 26] on both the background and mesh objects to avoid degenerate shapes—L2 on the Laplacian curvature [40], L1 on the angles between faces, and L1 on the variance in edge lengths.

**Implementation.** We implemented our model in TensorFlow [1]. For differentiable mesh rendering, we used the publicly-available code from [18, 19]; for voxel rendering, we implemented a custom renderer directly in TensorFlow. Hyperparameters were set by grid searches over blocks of related parameters. The values used in our final experiments are given in the supplementary material. We train each model on a single GPU, using gradient aggregation to increase the effective minibatch size. We train for approximately 100 epochs, which takes 2–5 days depending on the dataset.

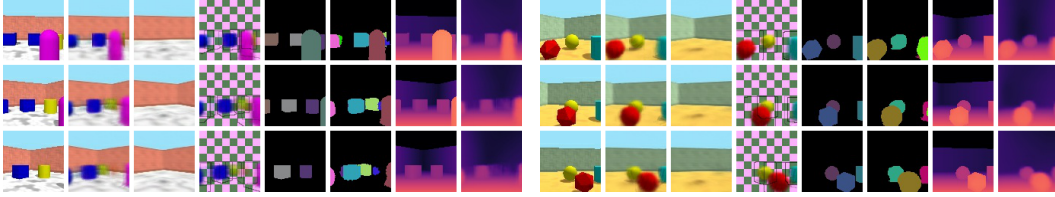

Figure 2: Videos from (ROOMS) decomposed by O3V-voxel. In each block, the three rows are different frames. The columns are (left to right): input frame, reconstructed frame, reconstructed background, reconstructed foreground objects and 3D bounding boxes, ground-truth and predicted instance segmentation, and ground-truth and predicted depth. Note the accurate segmentation and depth prediction, even when there is occlusion. Additional, larger examples are shown in the supplementary material.

# 4 Related work

We discuss three lines of work most relevant to ours—on generative models with explicit object structure, those which learn textured 3D shape from 2D data, and on object-aware video prediction.

**Object-centric generative models.** Several works propose generative models of images or videos, that explicitly reason about the objects they contain. AIR [13] uses spatial transformers to place isolated objects on a black background; SQAIR [29], R-SQAIR [41], DDPAE [23] and STOVE [30] extend this approach to videos such as bouncing digits. Other recent extensions focus on scalability and robustness to larger numbers of objects—SPAIR [8], GMIO [51], and SPACE [32] on images and SILOT [9] and SCALOR [24] on videos—this is orthogonal to our own focus on 3D scene representations. They treat objects as small 2D sprites, with associated locations and depth ordering, and compose them together to give images or videos; [2, 24, 32] also incorporate a 2D background. Like us, they use a large grid of candidate objects with presence indicators, but this grid tiles the 2D image instead of 3D space. These methods have only been applied to relatively simple datasets, where colors clearly delimit objects, and the background varies minimally over time. Moreover, [24, 32] treat objects as independent—meaning that they cannot generate coherent scenes. Another line of work treats objects as components of a spatial mixture model, explaining different regions of the image, but without considering their depths nor explicit locations [6, 12, 15–17]. Of these, only GENESIS [12] learns a prior that allows sampling coherent scenes. [45, 46] similarly use image-sized components for each object, but instead of a spatial mixture, they overlay them in depth order using latent binary masks.

**Unsupervised 3D generation.** Very recently, some works have attempted to learn generation of colored 3D shapes from only images. [20, 42] learn generative models over meshes, from image crops containing a single object of known class. However, these methods treat the background as 2D, and cannot synthesize complex scenes nor segment them into multiple objects; they also only operate on single images, hence are unable to exploit the richer information in videos. BlockGAN [35] and CIS [31] inject object-based 3D structure into generative adversarial networks, by placing blocks of features (ideally corresponding to individual objects) in 3D space, then projecting these to the image plane and mapping them to the final pixels with a CNN. This does not yield an *explicit* representation of 3D shape, does not guarantee disentanglement of object appearance from pose, and does not support inference of segmentation or 3D structure given an image.

**Object-aware video prediction.** Several works predict future video frames given an initial frame, considering the objects they contain. Note these models are *not* generative in the sense that ours is—they cannot sample entirely new videos. [50] predicts future appearance embeddings and 2D positions of objects, then decodes these to frames. [48] predicts frames by reasoning over object motion and background optical flow, but these are calculated in image space without considering the 3D scene. [36] does operate in 3D, but assumes depth images are given as input, and does not explicitly separate foreground objects from each other. These three methods require annotations specifying object locations. In contrast, [34] learns latent 2D keypoints whose motion is useful to explain future frames; similarly, [49, 52] discover latent instance segmentations and their dynamics.

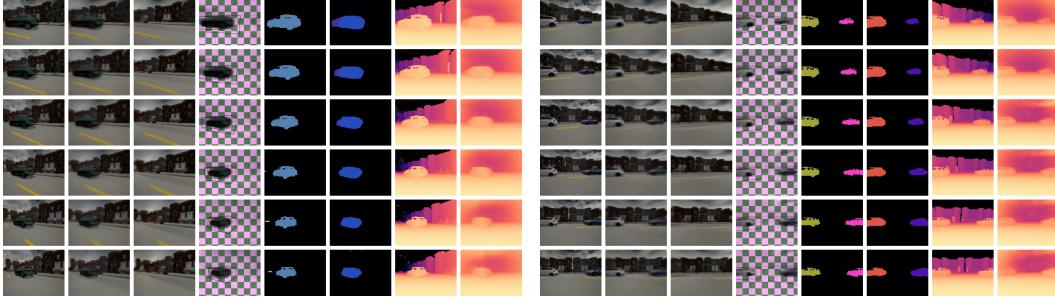

Figure 3: Videos from (TRAFFIC) decomposed by O3V-voxel (columns are as in Fig. 3). Cars are accurately segmented, though sometimes split into two parts. Depths are accurate in nearby areas, but less so in the far distance and sky. See the supplementary for more examples.

Table 1: Accuracy of prior works and the two variants of our method on foreground/background segmentation, per-frame instance segmentation, tracking instance segmentation, and 3D object detection. Dashes indicate a method cannot predict the required output.

| | (ROOMS) | | | | | | (TRAFFIC) | | | | | |
| | per-frame | | | tracking | | AP | per-frame | | | tracking | | AP |
| | fg-IOU | SC | mSC | SC | mSC | | fg-IOU | SC | mSC | SC | mSC | |
|---|---|---|---|---|---|---|---|---|---|---|---|---|
| MONet | – | 0.567 | 0.503 | – | – | – | – | 0.276 | 0.244 | – | – | – |
| GENESIS | – | 0.747 | 0.656 | – | – | – | – | 0.217 | 0.191 | – | – | – |
| SCALOR | **0.856** | **0.779** | **0.718** | 0.509 | 0.498 | – | 0.325 | 0.382 | 0.349 | 0.296 | 0.234 | – |
| O3V-voxel | 0.688 | 0.652 | 0.582 | **0.655** | **0.572** | **0.332** | **0.755** | **0.638** | **0.580** | **0.632** | **0.499** | 0.035 |
| O3V-mesh | 0.550 | 0.585 | 0.503 | 0.593 | 0.497 | 0.250 | 0.417 | 0.447 | 0.407 | 0.439 | 0.356 | **0.060** |

## 5  Experiments

We conduct experiments to validate that our model can predict the structure of 3D scenes shown in videos, including their decomposition into constituent objects. Then, we show that it can generate new, plausible videos from scratch. We name our model *O3V*, for Object-centric 3D Video modelling; we denote the variant with voxel objects by *O3V-voxel* and that with mesh objects by *O3V-mesh*. All reported results are means over four runs with different random seeds, disregarding runs that diverged early in training. The figures may be viewed as videos at http://www.pmh47.net/o3v/.

**Baselines.** We compare our performance to three state-of-the-art object-centric generative models: MONet [6], GENESIS [12] and SCALOR [24]. For MONet, we use the implementation from [12]; the other methods use the authors' original code. We tuned their hyperparameters for our datasets, e.g. the size and density of objects in [24], and the number of mixture components in [6, 12]. More details are given in the supplementary material.

**Datasets.** We use two datasets in our experiments, reserving 10% of each for validation and 10% for testing. The first, (ROOMS), is similar to the *rooms_ring_camera* dataset of [14], also used by [12].[3] This consists of 100K 3-frame sequences, each showing a room containing 1–4 static objects, with the camera positioned at a random location facing the center of the room, and moving on a circular path. As the original lacks 3D annotations, we generated our own version of this dataset; details are given in the supplementary material. The second dataset, (TRAFFIC), is generated using the CARLA driving simulator [10], which produces realistic videos of traffic scenes. Each sequence shows 1–3 cars driving in a street; the camera follows one car at varying angle. We rendered a total of 5000 80-frame sequences, and sampled random 6-frame sub-sequences to use as input for our model. Again, full details are given in the supplementary material. We emphasize that this dataset is considerably more challenging than any used to date for training unsupervised object-centric generative models, due to the complex foreground and background appearances, and camera motion.

Table 2: **(a)** Depth prediction accuracy on each dataset from our two model variants. Both work well for (ROOMS), where as O3V-voxel is better for (TRAFFIC). **(b)** Generation quality by our model and prior works. On video generation (FVD), our model is best for both datasets; on frame generation (FID/KID), GENESIS is better for the easier (ROOMS) dataset, but performs poorly on (TRAFFIC).

|  | (a) | | | | | (b) | | | | | |
| --- | --- | --- | --- | --- | --- | --- | --- | --- | --- | --- | --- |
|  | (ROOMS) | | (TRAFFIC) | | | (ROOMS) | | | (TRAFFIC) | | |
|  | MRE | $frac_{<1.25}$ | MRE | $frac_{<1.25}$ | | FVD | FID | KID | FVD | FID | KID |
| O3V-voxel | **0.058** | **0.949** | **0.201** | **0.661** | MONet | – | 158.0 | 0.151 | – | 265.2 | 0.305 |
| O3V-mesh | 0.067 | 0.943 | 1.260 | 0.312 | GENESIS | – | **96.9** | **0.083** | – | 199.7 | 0.257 |
|  | | | | | SCALOR | 1421.6 | 155.6 | 0.148 | 620.4 | 212.8 | 0.272 |
|  | | | | | O3V-voxel | **341.8** | 121.0 | 0.108 | **386.6** | **138.9** | **0.157** |
|  | | | | | O3V-mesh | 383.1 | 118.4 | 0.106 | 802.6 | 271.8 | 0.345 |

## 5.1 Scene decomposition

Our goal here is to measure how accurately our model decomposes videos into their constituent objects and background, and predicts the 3D structure of the scene.

**Metrics.** To measure how well foreground regions (i.e. objects) are segmented from background, we report the intersection-over-union between the true foreground mask and that predicted by the model (fg-IOU).[4] To measure how well individual object instances are segmented and tracked over time, we adopt the segmentation covering (SC) and mean segmentation covering (mSC) metrics of [12]. These match each ground-truth object to one predicted by the model, and evaluate their mean IOU [3]. SC weights objects according to their area, whereas mSC weights them equally. We use two variants—the *per-frame* metrics consider each frame individually (taking a mean over them); the *tracking* metrics consider the entire video jointly, so objects must be correctly tracked across frames to achieve a high score. Higher values are better.

Our model explicitly reconstructs videos in terms of a 3D background and objects; hence, it can predict a depth-map for each frame. We evaluate the accuracy of these using two standard metrics [11]. MRE is the mean relative error (lower is better), and $frac_{<1.25}$ is the fraction of pixels whose predicted depth is within a factor of $1.25$ of the true depth (higher is better). We also measure the 3D object detection accuracy, defined as the average precision (AP) with which predicted 3D bounding-boxes match the ground-truth; higher values are better.

**(ROOMS)** We see (Fig. 2) that our model faithfully reconstructs the input frames in all cases. Notably, the reconstructed depth-maps are also very similar to the ground-truth—although our method did not receive any 3D supervision. It correctly treats the walls and floor of the room as background, and the objects as foreground—even though the objects are stationary, meaning there are no motion cues. Quantitatively (Tab. 1), our method performs better than SCALOR on tracking instance segmentation, but slightly worse on the per-frame metrics. GENESIS also works well on the task it is designed for, but cannot distinguish foreground and background, nor track objects. With O3V-voxel, we achieve AP of 0.33 on 3D object detection—even though this is a challenging task where the model must accurately predict full 3D bounding boxes, and avoid multiple detections. O3V-voxel out-performs O3V-mesh on all metrics; we believe this is because meshes only yield gradients through pixels near the predicted object's surface, making it harder to escape local optima where object shapes and locations are poor. Tab. 2a shows that our model predicts depths accurately with both object representations, with a relative error of just 6%, and 94% of pixels having depth near the true value.

**(TRAFFIC)** Again, our model successfully reconstructs videos (Fig. 3); it assigns the road and buildings to the background, and cars to foreground objects. Interestingly, it sometimes segments the moving clouds as objects. Our evaluation metric penalizes this (as it regards only cars as foreground), but in fact it is semantically correct, as the clouds do move relative to the buildings. We see that sometimes a car is split into two parts; unfortunately this is a mathematically-correct solution subject to the constraints we impose on our problem and further regularization would be needed to avoid it. Quantitatively (Tab. 1), our method again performs better than SCALOR on tracking segmentation; it also out-performs MONet and GENESIS even on per-frame segmentation. This is notable since unlike on (ROOMS), color is not sufficient to infer object segmentation, and the model must in part base its segmentation on relative motion cues. Depth prediction is quantitatively poorer than

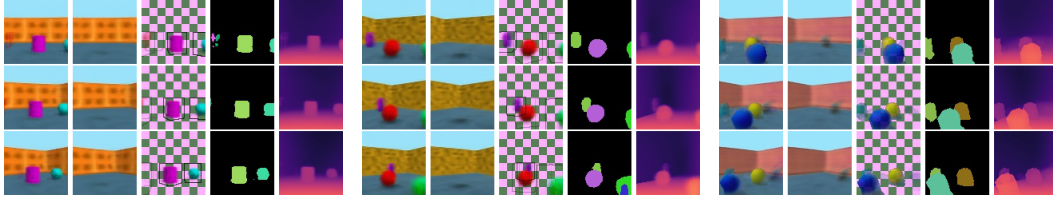

Figure 4: Videos generated by O3V-voxel (left two) and O3V-mesh (right two), trained on (ROOMS). In each block, the three rows are different frames. The columns are (left to right): generated frame, background, foreground objects with their 3D bounding boxes, instance segmentation, and depth. Note the diversity of wall/floor textures, and of object shapes, colors and locations.

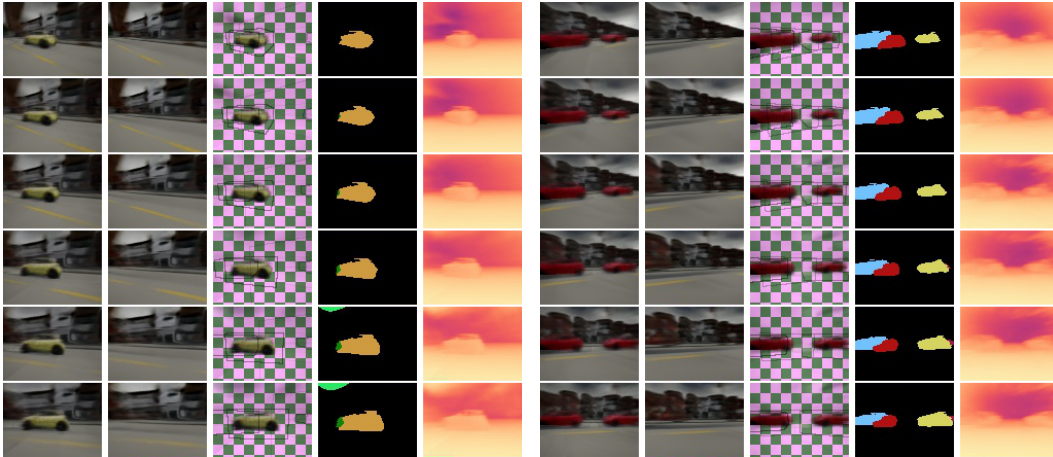

Figure 5: Videos generated by O3V-voxel trained on (TRAFFIC) (columns as in Fig. 4). The scenes are plausible and diverse, including the depth and segmentation. As in Fig. 3, some cars are split into two parts; however the two parts move together so the scene appears reasonable.

on (ROOMS), particularly when using O3V-mesh, however, the mean error is still only 20% for O3V-voxel, with 66% of pixels having depths close to correct. From Fig. 3, we see that cars and nearby road and buildings have generally accurate depths, while more distant parts of the background are foreshortened. Overall, we found O3V-mesh difficult to train on this dataset, as it was prone to diverge early in training. Nonetheless, on 3D object detection, O3V-mesh out-performs O3V-voxel, though neither method excels here. We emphasize however that our method is the first to tackle 3D detection with a generative model in the unsupervised setting.

## 5.2 Scene generation

Our goal here is to measure the quality of videos generated by our method—whether they are plausible, diverse, and resembling those in the training set.

**Metrics.** We adopt the Fréchet video distance (FVD) metric of [44] to measure how similar the distribution of our generated videos is to the test set. We also measure Fréchet Inception distance (FID) [21] and Kernel Inception distance (KID) [4], treating frames independently, to compare with methods that only generate images, not videos. These metrics measure the divergence between

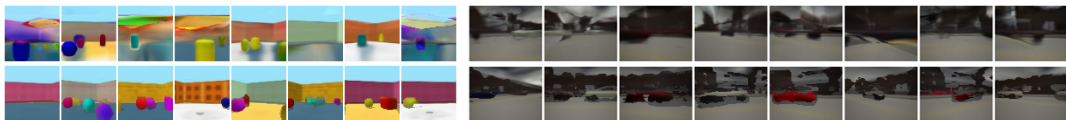

Figure 6: Single frames generated by MONet [6] (top) and GENESIS [12] (bottom) trained on our datasets. Further examples are given in the supplementary.

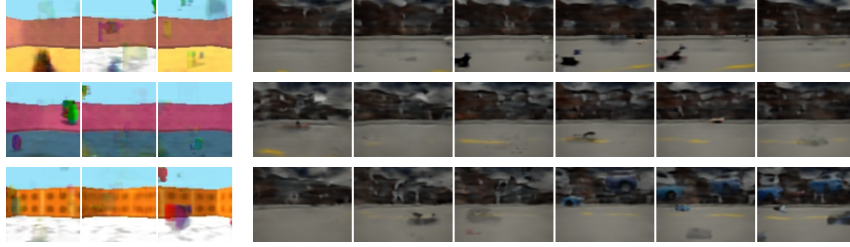

Figure 7: Videos generated by SCALOR [24] trained on our datasets. Each row is a different episode. While the model can generate plausible backgrounds, the foreground objects are fragmentary and incoherent. Further examples are given in the supplementary.

distributions of feature activations in pretrained CNNs [7, 43] when generated and ground-truth samples are passed to them. Lower values are better.

**(ROOMS)** Our method generates coherent videos (Fig. 4), showing varying numbers of objects in rooms resembling those in the training set. The textures are crisp and detailed, and the model successfully mimics shading variations on the objects. As the camera moves, the view remains plausible, with objects correctly occluding one another, due to our explicitly endowing the model with 3D structure. The scene-level prior ensures that objects are correctly spaced out across the floor, and do not intersect each other nor the walls. For O3V-mesh, there is often a small 'foot' on objects, where the area of the floor darkened by the object's shadow is included as part of the object itself. An interesting failure case is the rightmost sequence, where a small, distant object is 'painted' onto the background. Samples generated by the baselines are displayed in the left columns of Fig. 6 and Fig. 7. Qualitatively, and quantitatively according to FVD and FID/KID (Tab. 2b), our method significantly out-performs SCALOR; similarly it out-performs MONet according to FID and KID and visual quality. We attribute this to the scene-level prior, which allows sampling coherent scenes, rather than just fragments. GENESIS, which does incorporate such a prior, also performs better—even better than our method according to FID and KID—but is limited to generating isolated frames, not videos. Visually, its generations appear to be of similar quality to ours. We attribute this good performance to the clear segmentation of objects by color on this dataset. Interestingly, O3V-mesh achieves better FID and KID than O3V-voxel—perhaps because meshes naturally yield sharp edges when rendered, resulting in images with local texture statistics closer to the originals.

**(TRAFFIC)** On this more challenging dataset, O3V-voxel significantly out-performs all other methods in terms of FVD, FID and KID. GENESIS is second-best, slightly beating O3V-mesh; we believe its lower performance here than on (ROOMS) is due to color being a weaker indicator of object segmentation. In accordance with the decomposition results, we see O3V-mesh is significantly worse than O3V-voxel. Qualitatively, we see (Fig. 5) that the results again typically show coherent scenes, with one or two cars and a plausible background. Moreover, the model has learnt that cars move along the line of the road (which varies in the ego-centric frame-of-reference). However, as noted in the decomposition results, the model sometimes splits a car into two parts. When this occurs, the two parts often move as one; in the supplementary we highlight some failure cases where separated half-cars are sampled. Samples generated by the baselines are displayed in the right columns of Fig. 6 and Fig. 7; these accord with the quantitative results. In particular, we see that GENESIS again out-performs MONet and SCALOR in terms of frame quality, but its samples are significantly lower-quality than O3V-voxel (Fig. 5). SCALOR produces reasonable backgrounds, but its inability to learn a scene-level prior means the foreground objects are fragmentary and incoherent.

## 6 Conclusion

We have presented a new object-centric generative model of videos that explicitly reasons in 3D space, and shown it successfully decomposes scenes into 3D objects, predicts depth and segmentation, and generates plausible samples. Our innovations are orthogonal to recent work on attentive inference models [6, 15], scalability [9, 24, 32], and broadcast decoders [6, 12], and it would be interesting future work to investigate combining them. In particular, a compositional encoder would complement our compositional generative process, and hopefully allow training on videos with a larger number of objects.

## Broader Impact

This paper has tackled several tasks—segmentation, tracking, and 3D detection—that have been the subject of extensive research in computer vision, and have well-known societal implications. These include both positive impacts such as life-saving self-driving cars and safety systems, and negative impacts such as potentially-intrusive surveillance technologies. However, currently-deployed approaches to these tasks rely on *supervised* learning, in a discriminative setting—whereas we consider the unsupervised, generative setting. This limits the immediate impact of the present work, particularly given we only consider synthetic imagery—though one of our contributions is to narrow the gap between the complexity of videos that are handled by unsupervised and supervised methods. If unsupervised methods do become competitive with fully-supervised ones or are applied more generally, it is as yet unclear what the implications will be. Supervised learning is known for 'transferring' bias from human annotators to the model; unsupervised methods should avoid this problem, but any biases that do arise are likely to be even harder to diagnose. Of course, reducing reliance on human annotators is itself a non-trivial societal impact—most directly due to loss of annotator jobs, but also by enabling new applications for which human annotation remains too costly. In particular, unsupervised techniques should allow leveraging the enormous volume of video content that already exists, at minimal cost in labor. A potential negative impact of all generative models, is that they can be applied to the creation of fake content—by advancing the quality of such models, we also indirectly aid those who create such fakes. Conversely, we also aid those who use generative models for artistic and other positive purposes, and we hope these benefits outweigh the downsides.

## Acknowledgments and Disclosure of Funding

This research was supported by the Scientific Service Units (SSU) of IST Austria through resources provided by Scientific Computing (SciComp). PH is employed part-time by Blackford Analysis, but they did not support this project in any way.

## Footnotes

[1]All network architectures are given in the supplementary material, and code is available at `https://github.com/pmh47/o3v`

[2]We constrain object motion and rotations to lie in the $xz$-plane, which is sufficient for the datasets we use

[3][32] used a similar dataset with different statistics (more objects but simpler textures)

[4]Further details on all the metrics are given in the supplementary material

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
