[Supplementary Material · supplementary.pdf]

# Supplementary Material for
## *Unsupervised object-centric video generation and decomposition in 3D*

## S1  Baselines

In this section, we discuss our baseline experiments using MONet [3], GENE-SIS [7] and SCALOR [12] in more detail. For MONet, we used the reimple-mentation from the authors of [7]; note that this uses a different loss compared with the original. For GENESIS and SCALOR, we used the authors' original implementations. These methods operate on images of size $64 \times 64$. Rather than modifying their architectures to suit our larger images, we instead downscale our images, using bicubic interpolation. We also tuned their hyperparameters—for MONet and GENESIS, the number of mixture components, and for SCALOR, the number of grid cells, and ranges of the object size and aspect ratio param-eters. We trained MONet and GENESIS for 2M iterations, and SCALOR for 1.5M iterations on (ROOMS) and 500K iterations on (TRAFFIC).

**Qualitative results.**  We show examples of frames/videos generated by MONet (Figure S1 and Figure S2), GENESIS (Figure S3 and Figure S4) and SCALOR (Figure S5 and Figure S6) trained on our datasets. We see that GENESIS achieves high-quality generations on (ROOMS), surpassing even our method—but, it can only generate isolated frames, not complete videos. MONet also generates only frames, and moreover, its lack of a scene-level prior causes them to be fragmentary and incoherent (as also noted by [7]). SCALOR can generate full videos, and these often have reasonable backgrounds—but again, the lack of prior on foreground objects results in implausible appearances. On (TRAFFIC), all three methods produce slightly poorer results, but GENESIS again yields the best samples.

## S2  Datasets

### S2.1  (ROOMS)

This dataset is based on the *rooms_ring_camera* dataset of [8]. Their dataset is publicly-available, but does not include 3D annotations, which are important

Figure S1: Frames generated by MONet [3], trained on our dataset (ROOMS)

Figure S2: Frames generated by MONet [3], trained on our dataset (TRAFFIC)

Figure S3: Frames generated by GENESIS [7], trained on our dataset (ROOMS)

Figure S4: Frames generated by GENESIS [7], trained on our dataset (TRAFFIC)

Figure S5: Videos generated by SCALOR [12], trained on our dataset (ROOMS). Each column is a different episode. While the model can generate plausible backgrounds, the foreground objects are fragmentary and incoherent.

Figure S6: Videos generated by SCALOR [12], trained on our dataset (TRAF-FIC). Each column is a different episode. While the model can generate plausible backgrounds, the foreground objects are fragmentary and incoherent.

to validate our method. We therefore generated a similar dataset ourselves, by adapting the code at `https://github.com/musyoku/gqn-dataset-renderer`. We record short, six-frame videos; each shows a room of fixed size, with 1–4 objects placed at random locations, without intersections. There are five textures for the walls and three for the floor; these are randomly selected. Each object has one of five different shapes (cube, sphere, capsule, cylinder, icosahedron) and six colors, sampled independently. The camera starts at a random azimuth and distance from the center of the room, then moves around it at constant rate, rotating so it always faces the center. As input to our model, we randomly sample contiguous three-frame sub-sequences from each six-frame video. Note that because the objects are static, shorter videos make for a harder learning problem, as the 3D structure is more ambiguous. For more details and sampling parameters, see the publicly-available code.

## S2.2 (TRAFFIC)

This dataset is generated using the CARLA driving simulator [5], which produces realistic videos of road scenes, using detailed models and physically-principled shading. We use their map *Town02*, and create 4–5 cars of random models and colors, at random positions along a road. The cars then move according to simulated driving, with different speeds; however, we manually increase the probability that multiple cars follow each other closely (to increase the number of episodes with more than one object). The camera follows one of the cars (chosen at random), moving on an elliptical path around it, at random radius and speed. We capture frames at 4 FPS, taking around 80 frames per video; we extract contiguous six-frame sub-sequences to use as input to our model. The simulator outputs only semantic segmentations, not the instance segmentations we require for evaluation. To construct instance segmentations, we project the ground-truth 3D bounding-boxes back into the frames, and intersect these with the semantic segmentation mask for 'car'. Each pixel of the 'car' mask is only allowed to belong to one instance, with the nearer one taking precedence. Visually, this gives a very accurate instance segmentations for the majority of cases; however, there are occasionally slight errors when one car strongly occludes another. For more details and sampling parameters, see the publicly-available code.

# S3 Evaluation metrics

## S3.1 Segmentation

We report five segmentation metrics. These all operate on binary masks, which we generate from our reconstructed scenes. For O3V-voxel, we render each object individually over a black background with its color set to white, but using the predicted presence and opacity values. This yields a soft, amodal segmentation mask for each object and each frame. To produce a single, modal

instance segmentation for a frame, we binarize these by thresholding them, then stack the resulting masks in depth order, allowing nearer object masks to occlude farther ones. For O3V-mesh, we render the silhouette of each object, and stack these in depth order, discarding any with presence below a threshold. Our metrics are then calculated based on the resulting instance segmentations.

The foreground intersection-over-union (fg-IOU) considers only the assignment of pixels to foreground versus background, regardless of object identity. For (ROOMS), we treat the objects as foreground, and the room (walls and floor) and sky as background. For (TRAFFIC), we treat cars as foreground, and everything else as background. We calculate the IOU per-frame, then take a mean over frames.

The segmentation covering (SC) and mean segmentation covering (mSC) metrics were proposed by [7]. These match each ground-truth foreground object to one predicted by the model, and evaluate their mean IOU [1]. SC weights objects according to their area, whereas mSC weights them equally. Unlike fg-IOU, a method must correctly separate individual object instances to score highly. Our first variant of these metrics, *per-frame*, treats frames independently (taking a mean over them). We refer the reader to the calculations described in [7]; we use the inferred instance segmentation described above as the predicted input. Our second variant requires objects to be correctly tracked over time; we achieve this by using the same metric, but treating the entire video as a single large image—so the IOU calculation reduces over time as well as space.

## S3.2    Depth prediction

We report two metrics for depth prediction, used by [6] and numerous earlier works on classical stereo depth prediction. MRE is the mean absolute relative error in predicted depths. $\text{frac}_{<1.25}$ is the fraction of pixels whose predicted depth is within a factor of 1.25 of the true depth; this particular factor is chosen because it reflects the accuracy with which humans can judge depths in images. We also implemented several other similar metrics, but found them to be well-correlated with those we report.

These metrics both require a predicted depth-map as input. However, our reconstructions contain partially-opaque objects when the presence indicators are not exactly zero or one. We therefore render depth-maps for the background and each object, then combine these by alpha-blending according to the object presences—the same as when we render the final frames.

## S3.3    3D object detection

To measure the quality of object detection, we use a variant of the 3D AP metric from the KITTI dataset [10]. We assign a 3D bounding box and a score to each predicted object in each frame, then match these to ground-truth bounding boxes, penalizing multiple detections of the same instance. Specifically, for O3V-voxel, for each object and frame, we find the axis-aligned bounding box in view-space (AABB) of those voxels with higher than threshold opacity; the

score is set to the maximum opacity multiplied by the presence indicator. For O3V-mesh, we similarly find the AABB of the vertices, and set the score equal to the presence indicator. Then, for each ground-truth object, we find the most-overlapping predicted object with IOU > 0.3 (i.e. roughly 2/3 overlap along each axis) and define this as a true-positive; all others (included repeated detections of the same ground-truth) are false-positives. The true-positives and false-positives are ranked by the scores, and the area under the interpolated precision-recall curve calculated to give the average precision (AP) [9].

## S3.4   Video generation

Evaluating image and video generation is notoriously difficult. We adopt three standard metrics that aim to measure how close a distribution of generated videos is to that of (held-out) ground-truth videos:

- Fréchet video distance (FVD) [13] measures similarity of the logits of the I3D action-recognition network [4]. The architecture of I3D limits the minimum supported video length; where our videos are too short, we extend them by 'ping-ponging', i.e. concatenating the video with its reverse. This avoids discontinuities caused by simply repeating the frames from the start.

- Fréchet Inception distance (FID) [11] is more widely adopted than FVD, but considers images rather than videos. We therefore apply it independently to each frame, following the standard protocol, then take a mean over frames.

- Kernel Inception distance (KID) [2] is an alternative to FID, that has a simple unbiased estimator. Again we apply it per-frame and take the mean.

# S4 Hyperparameters

| | (ROOMS) | | (TRAFFIC) | |
| --- | --- | --- | --- | --- |
| | O3V-voxel | O3V-mesh | O3V-voxel | O3V-mesh |
| **Model** | | | | |
| scene dimensionality $d$ | 32 | 32 | 64 | 64 |
| camera dimensionality $c$ | 128 | 128 | 128 | 128 |
| object dimensionality $e$ | 16 | 16 | 16 | 16 |
| background vertices $N \times M$ | $24 \times 64$ | $24 \times 64$ | $24 \times 64$ | $24 \times 64$ |
| background offset scale $\gamma$ | 1 | 1 | 1 | 1 |
| grid dimensions $G$ | $6 \times 1 \times 7$ | $6 \times 1 \times 7$ | $5 \times 1 \times 7$ | $5 \times 1 \times 7$ |
| object vertices $S \times T$ | – | $8 \times 16$ | – | $8 \times 16$ |
| object offset scale $\gamma$ | – | 0.2 | – | 0.2 |
| object velocity bias $\hat{\mathbf{v}}$ | $(0,0,0)$ | $(0,0,0)$ | $(-1,0,0)$ | $(-1,0,0)$ |
| **Loss** | | | | |
| scale-space pyramid depth | 4 | 5 | 4 | 5 |
| initial KL weight $\beta$ | 1 | 0.5 | 0.5 | 0.5 |
| final KL weight $\beta$ | 1 | 2 | 2 | 0.5 |
| L1 velocity strength | 1 | 1 | 1 | 1 |
| presence hinge strength | – | 100 | – | 10 |
| L2 Laplacian strength (obj) | – | 7.5 | – | 100 |
| L1 crease strength (bg) | 10 | 10 | 50 | 50 |
| L1 edge-length variance (bg) | 10 | 10 | 10 | 10 |
| edge-matching strength | 0 | 0 | 10 | 0 |
| edge-matching $\zeta$ | – | – | 10 | – |
| **Optimization** | | | | |
| batch size | 64 | 32 | 12 | 12 |
| learning rate | $10^{-4}$ | $10^{-4}$ | $10^{-4}$ | $10^{-4}$ |

# S5 Network architectures

This specifies describes the encoder and decoder network architectures for each component of our model. Unspecified parameters are assumed to take Keras defaults.

## S5.1 Video encoder

Conv3D(32, kernel size=[1, 7, 7], strides=[1, 2, 2], activation=relu)
GroupNormalization(groups=4)
Conv3D(64, kernel size=[1, 3, 3], strides=[1, 2, 2], activation=relu)
Residual(Conv3D(64, kernel_size=[1, 3, 3], activation=relu, padding=SAME))
GroupNormalization(groups=4)
Conv3D(96, kernel size=[2, 1, 1], activation=relu)
GroupNormalization(groups=6)
Conv3D(128, kernel size=[1, 3, 3], strides=[1, 2, 2], activation=relu)
Residual(Conv3D(128, kernel size=[1, 3, 3], activation=relu, padding=SAME))
GroupNormalization(groups=4)
Conv3D(192, kernel size=[2, 1, 1], activation=relu)
GroupNormalization(groups=6)
Conv3D(256, kernel size=[1, 3, 3], activation=relu)
Flatten
LayerNormalization
Dense(1024, activation=relu)
Residual(Dense(activation=relu))

## S5.2 Camera parameter encoder $F^{\mathrm{cam}}$

Dense(128, activation=elu)
LayerNormalization
Residual(Dense(activation=elu))
LayerNormalization
Residual(Dense(activation=elu))
LayerNormalization

## S5.3 Background decoder (shape) $D^{\mathrm{bg}}_{\mathrm{shape}}$

The first three layers are shared with $D^{\mathrm{bg}}_{\mathrm{tex}}$.

Dense(128, activation=elu)
Residual(Dense(activation=elu))
Dense(12)
Dense(4)
Dense(480, activation=elu)
Reshape([3, 8, -1])
Conv2D(96, kernel size=[3, 3], activation=elu, padding=SAME)
Residual(Conv2D(filters=96, kernel size=1, activation=elu))
UpSampling2D
Conv2D(64, kernel size=[3, 3], activation=elu, padding=SAME)
Residual(Conv2D(filters=64, kernel size=1, activation=elu))
UpSampling2D
Conv2D(48, kernel size=[3, 3], activation=elu, padding=SAME)
Residual(Conv2D(filters=48, kernel size=1, activation=elu))
UpSampling2D
Conv2D(32, kernel size=[3, 3], activation=elu, padding=SAME)
Residual(Conv2D(filters=32, kernel size=1, activation=elu))
Conv2D(4, kernel size=[3, 3], padding=SAME)

## S5.4  Background decoder (texture) $D_{\text{tex}}^{\text{bg}}$

The first three layers are shared with $D_{\text{shape}}^{\text{bg}}$.

Dense(128, activation=elu)
Residual(Dense(activation=elu))
Dense(12)
Dense(64)
Dense(720, activation=elu)
Reshape([6, 12, -1])
Conv2D(128, kernel size=[3, 3], activation=elu, padding=SAME)
Residual(Conv2D(filters=128, kernel size=1, activation=elu))
UpSampling2D
Conv2D(96, kernel size=[3, 3], activation=elu, padding=SAME)
Residual(Conv2D(filters=96, kernel size=1, activation=elu))
UpSampling2D
Conv2D(64, kernel size=[3, 3], activation=elu, padding=SAME)
Residual(Conv2D(filters=64, kernel size=1, activation=elu))
UpSampling2D
Conv2D(48, kernel size=[3, 3], activation=elu, padding=SAME)
Residual(Conv2D(filters=48, kernel size=1, activation=elu))
UpSampling2D
Conv2D(32, kernel size=[3, 3], activation=elu, padding=SAME)
Residual(Conv2D(filters=32, kernel size=1, activation=elu))
UpSampling2D
Conv2D(24, kernel size=[3, 3], activation=elu, padding=SAME)
Residual(Conv2D(filters=24, kernel size=1, activation=elu))
Conv2D(3, kernel size=[3, 3])

## S5.5   Object parameter decoder $F^{\mathrm{obj}}$

Dense(128, activation=elu)
Residual(Dense(activation=elu))
Dense($G \times (e + 8 + 2(L - 1))$)

## S5.6 Object appearance decoder (voxels) $D^{\mathrm{obj}}_{\mathrm{voxels}}$

Dense($3 \times 3 \times 64$, activation=elu)
Reshape([3, 3, 64])
UpSampling2D
Conv2D(64, kernel size=3, padding=SAME, activation=elu)
UpSampling2D
Conv2D(48, kernel size=3, padding=SAME, activation=elu)
UpSampling2D
Conv2D(32, kernel size=3, padding=SAME, activation=elu)
Conv2D($24 \times 4$, kernel size=4, padding=SAME, activation=None)
Reshape([24, 24, 24, 4])
Permute([3, 1, 2, 4])

## S5.7 Object appearance decoder (mesh shape) $D^{\mathrm{obj}}_{\mathrm{shape}}$

Dense(128, activation=elu)
Reshape([2, 4, -1])
Conv2D(96, kernel size=[2, 2], activation=elu, padding=SAME)
UpSampling2D
Conv2D(64, kernel size=[3, 3], activation=elu, padding=SAME)
Residual(Conv2D(filters=64, kernel size=1, activation=elu))
UpSampling2D
Conv2D(48, kernel size=[3, 3], activation=elu, padding=SAME)
Conv2D(4, kernel size=1, kernel initializer=ZEROS)

## S5.8 Object appearance decoder (mesh texture) $D^{\mathrm{obj}}_{\mathrm{tex}}$

Dense(180, activation=elu)
Reshape([3, 6, -1])
Conv2D(96, kernel size=[2, 2], activation=elu, padding=SAME)
Residual(Conv2D(filters=96, kernel size=1, activation=elu))
UpSampling2D
Conv2D(64, kernel size=[3, 3], activation=elu, padding=SAME)
Residual(Conv2D(filters=64, kernel size=1, activation=elu))
UpSampling2D(size=4, interpolation=BILINEAR)
Conv2D(24, kernel size=[3, 3], activation=elu, padding=SAME)
Conv2D(3, kernel size=1)

## S6    Longer sequences

In this section, we present qualitative results from O3V-voxel when run on significantly longer sequences for (ROOMS)—12 frames instead of three frames. To control memory usage so the model can still be trained on a single GPU, we only render six frames per training episode—however we unroll the full 12 steps in latent space for all sequences, and generate/reconstruct full sequences at test time. We also add an extra downsampling layer at the start of the encoder CNN. This dataset required slightly modified hyperparameters; in particular, we found it beneficial to adjust the translation and scale of the background vertices. Without these changes, the larger range of camera motion in the longer sequences results in undesirable local optima where the camera passes through the background surface early in training. This is not recoverable by the optimisation process, as the camera passing through the surface produces a discontinuous change in the pixels, so there is no gradient signal.

    Generation results are shown in Figure S7; decomposition in Figure S8. We see that the model still performs reasonably well in this more-challenging setting. In particular, the reconstructions are faithful, with good foreground-background segmentation. However, the depth-maps indicate that the corners between walls are less crisply predicted than for shorter sequences. Generations suffer from some blurring in the background texture, but the scenes are still coherent, and most objects recognisable.

## S7    Additional qualitative results

In this section, we present further qualitative results from our method. These figures are extended versions of those in the main paper, with larger images and more examples. Videos of the same sequences may be seen at `http://pmh47.net/o3v/`. Figure S9, Figure S10, Figure S11 and Figure S12 show decomposition results. Figure S13, Figure S14, Figure S15, Figure S16 and Figure S17 show generated samples. Figure S18 shows selected examples on (TRAFFIC) where our model fails, and samples independently-moving pieces of cars.

Figure S7: Generation results from O3V-voxel for 12-frame sequences from (ROOMS). Each group of three rows shows one video; within each group, the rows are generated frames, generated foreground only, and depth-maps.

Figure S8: Decomposition results from O3V-voxel for 12-frame sequences from (ROOMS). Each group of four rows shows one video; within each group, the rows are original frames, reconstructed frames, reconstructed foreground only, and predicted depth-maps.

Figure S9: Videos from (ROOMS) decomposed by O3V-voxel. In each block, the three rows are different frames. The columns are (left to right): input frame, reconstructed frame, reconstructed background, reconstructed foreground objects and **3D bounding boxes**, ground-truth and predicted instance segmentation, and ground-truth and predicted depth.

Figure S10: Videos from (ROOMS) decomposed by O3V-mesh. In each block, the three rows are different frames. The columns are (left to right): input frame, reconstructed frame, reconstructed background, reconstructed foreground objects and 3D bounding boxes, ground-truth and predicted instance segmentation, and ground-truth and predicted depth.

Figure S11: Videos from (TRAFFIC) decomposed by O3V-voxel (columns are as in Figure S9). Cars are accurately segmented, though sometimes split into two parts. Depths are accurate in nearby areas, but less so in the far distance and sky.

Figure S12: Continuation of Figure S11

Figure S13: Videos generated by O3V-voxel trained on (ROOMS). In each block, the three rows are different frames. The columns are (left to right): generated frame, background, foreground objects with their 3D bounding boxes, instance segmentation, and depth.

Figure S14: Videos generated by O3V-mesh trained on (ROOMS). In each block, the three rows are different frames. The columns are (left to right): generated frame, background, foreground objects with their 3D bounding boxes, instance segmentation, and depth.

Figure S15: Videos generated by O3V-voxel trained on (TRAFFIC) (columns as in Figure S13).

Figure S16: Continuation of Figure S15

Figure S17: Continuation of Figure S15

Figure S18: Selected examples of failures from O3V-voxel trained on (TRAFFIC) (columns as in Figure S13). In all these examples, our method incorrectly samples several independently-moving partial cars.