[Reviews · NeurIPS 2020]

Review 1

Summary and Contributions: [Main Contributions] - Generating coherent short videos of 3D scenes. Significance: high - Learning 3D scene representations for objects and background in an unsupervised fashion using 2D images with a compositional generative model. Significance: medium Reviewer’s legend for the level of significance: [very low, low, medium, high, very high] The ratings are a function of the results/analysis shown in the paper, current state of the art methods and the reviewer’s perspective on the authors’ approach [High-level summary of the paper] This paper proposes a generative model for videos which operates over disentangled 3D representations of objects and a 3D background, without direct 3D supervision. They show that their models enables them to generate videos of scenes with objects in it by moving the camera. Through amortized inference, it is also possible to decompose a scene into its constituent 3D objects and a 3D background. [A bit more low-level summary] The proposed model decomposes a scene into multiple 3D foreground objects and a 3D background in a VAE-like framework whose decoder consists of a neural network component and a non-neural network differentiable renderer. During inference, [3, 6] frames of a video are encoded into a d-dimentional Gaussian latent variable Z. Then the Z embedding is decoded into object representations for each object which include appearance properties in addition to time-varying 3D position and pose. The neural network decoder operates on the appearance properties and produces a 3D shape (as mesh or voxel) with a texture. Then a differentiable renderer takes in the background (a mesh) and each of the produced 3D shapes along with the time-varying 3D position and pose and renders the final 2D image. The authors compare their method against recent and state-of-the-art methods (MONet, GENESIS and SCALOR) and show that their method performs better for both scene decomposition and generation of videos.

Strengths: [Paper strengths] - The paper’s organization is good and each section is well-developed. The high-level motivations of the paper are of interest to the NeurIPS community. - The proposed method seems to be able to learn the distribution of the scenes and generate scenes that seem realistic with respect to the data manifold. This work is a good proof-of-concept in terms of showing the importance of the inductive biases that the authors have baked into their model such as the 3D object and background representations and compositionality Technical strengths: - The authors propose a model that is able to efficiently encode/decompose scenes containing up to 4 objects into 3D representations - The experiments show that the authors’ proposed approach works better than prior works (although prior works were not designed to do the same task) - The model is trained using 2D images only, hence no direct 3D supervision

Weaknesses: [Paper weaknesses] - Despite that the contributions of the work are good but the results do not meet the expectations set by paper’s tone and the claims of the authors make (see below for more details on this) Technical weaknesses: - Videos are low-resolution and visual fidelity is low, specially given that most shapes are simple - The rendering of the scenes deteriorates as the number of objects goes beyond 2. This probably means that the object representations do not carry all the necessary information or the model does not seem to be leveraging the compositionality inductive bias well. This could partly be resolved by not having to encode the all video frames into a single latent code; an attention mechanism during inference and a DRAW-like generative model (Gregor et al.) might help with this. - The authors do not conduct any experiment to evaluate their model’s out of distribution generalization capability. The experiments can include decomposing a scene with a new background and object color/texture and also generating scenes with more than 4 objects. The current modeling paradigm might be a limiting factor to allow generating scenes with more objects though. - The camera is always positioned the same for all videos which makes gives the impression that the model has not properly learned 3D scene representations and gives the reader the impression that that the rendering output could look pretty bad if the camera would move to another position. - The authors do not show scene generation results from the prior works. The F[I/V]D metric is not unbiased and accurate and cannot be used as a determinant factor for which model generates better results. The authors may consider using the Kernel Inception Distance (for reference take a look at the paper “Demystifying MMD GANs”) metric in their evaluations. - This is not very important but it would have been be interesting if the authors could have shown renderings when they remove some of the inductive biases in their model, specially the inductive biases for the objects. This would help the reader and the community to see the contribution of incorporating such inductive biases. - I know this is generally difficult to train any model to handle videos but the videos used to train the model are really short (3 or 6 frames).

Correctness: [Claims] Reading the abstract and introduction raises the expectations of the reader but the results and later parts of the paper that describe more details of the model show a mismatch between the claims implied/made and the things the paper delivers. See some examples below. - Unsupervised learning only from monocular images: [partially/mostly] correct. It is correct that the main training objective is based on the signal that comes from the 2D images. However, it seems that although the inductive biases baked into the model (specially the carefully-designed disentangled objects and background representations) are potentially making the learning much easier, by providing indirect 3D supervision. The regularization might be helping to some degree as well. - Generate coherent videos: partially/mostly correct. The resolution of the videos is low, the videos are very short and the visual fidelity is usually low as well which makes it hard to judge. It is likely that longer videos could not be coherent. - The authors claim that disentanglement of pose and appearance and decoding objects one at a time allows handling videos with high visual complexity: not correct. The model does not seem to be able to generate/decompose scenes with more than 4 objects. Additionally, the scenes are simple in general.

Clarity: The paper is written well and the authors have done a good job to developed each section. The supplementary materials contains additional results; thanks to the authors for this.

Relation to Prior Work: Overall, the authors have done a good job to review prior works and describe their shortcomings in addition to how their work has tried to address them.

Reproducibility: Yes

Additional Feedback: [Additional Comments] Introduction: - The introduction is written well overall. - Lines 24-25: The fact that the model can decompose a scene into foreground and background objects is mainly due to design choices of the authors that makes it hard for the model to choose to use the background mesh to represent the objects. Though I understand that learning this automatically is a hard problem. Method: - The method section is well-developed and the supplementary materials provide further details/results in addition to implementation details. - Section 2.2: The inductive biases the authors mention here make the problem of learning videos much simpler. It would be much more interesting if the model was automatically learning most or all of these parameters in a disentangled fashion. - For the tables please make the numbers that correspond to better values (lower loss/higher accuracy) bold so that it would be easy to recognize which method performs better. [Rebuttal (prioritized)] - I know it is a hard problem to generate long videos. However, I think the authors should have done an experiment to show that the model has learned a good 3D representation of the scene. For this it would be good if the authors can produce short videos of a scene where the video begins with the camera start to move from multiple, different, locations in the same scene, instead of one fixed position as shown in the results. This will help me see how the scene is represented from the model’s perspective. It would be helpful if the authors can do this for multiple scenes and for both decomposition and generation tasks (e.g. 3 scenes decomposed and 3 generated scenes) - I’m curious to see videos of generated scenes from of the model when trained only on single images of each scene instead of 3 or 6. - Lines 86-87: I am not sure what the authors mean by “this grid covers the 3D space of the scene, rather than 2D pixel space”. Also, as far as I know, in reference 34 there is no “grid of candidate objects” - I know that Generative Query Networks (Eslami et al.) does not have a compositional and an explicit object-centric representation but it seems that GQNs is in practice might be able to do most of what the authors have done here without the additional inductive biases that the authors have used in their work. Why the authors have not compared their work to GQNs for scene decomposition and generation experiments? - On a related note, it would be good if the authors can train a Scene Representation Networks (Sitzmann et al.) on their data and compare it to their model.


Review 2

Summary and Contributions: This paper proposes a model for unsupervised object-centric video generation and decomposition in 3D. The proposed model decomposes a latent code into a background, and foreground objects at different locations in a 3D grid, and their velocities (linear and angular). This model is trained without any direct supervision but by maximizing the likelihood of videos using variational training (maximizing the ELBO). This requires differentiable rendering of the latent code into an image. Paper investigates two variants for doing this: rendering meshes and rendering voxel grids. Paper additionally imposes regularization terms to avoid degenerate solutions. Paper evaluates the proposed model by decomposing videos into constituent objects, determining its 3D structure, and generating coherent videos.

Strengths: 1. I found the part of the formulation that generates an object at each location (along with a probability if the object is there or not), pretty neat. I believe this could allow the model to generalize across number of object instances, though the paper does not empirically show this. 2. The proposed model makes sense. 3. Empirical evaluation tests various aspects of the proposed model. Model compares favorably with recent methods for related tasks.

Weaknesses: 1. While proposed method seems to perform favorably as compared to past methods, empirical evaluation has only been done on synthetic videos. Synthetic videos are in many ways much simpler than real videos (in terms of complexity with respect to number of objects, lighting, appearance and material properties). While it is great that the model works well in simulated environments (it is a hard problem), I strongly suspect that the model will perform very poorly on real data. It will strengthen the paper if some experiments were conducted on real data. 2. It is not clear to me, as to what is the precise novelty. The proposed model is a combination of known techniques in the area: differentiable rendering of meshes and voxel grids, convolutional prediction of objects in 3D (3D versions of Faster RCNN). I agree that the combined system is new, and the demonstration that it works (in simulated videos) is great. However, I am not sure what I learn from this paper. 3. Additional concerns about experiments: a) why does FID based on real image networks be a good measure for image quality, given all images here are synthetic? b) why are trends very different between rooms and traffic datasets (per-frame metrics with respect to SCALOR, ordering of O3D-voxel and O3D-mesh for FID between traffic and room). 4. Finally, I am wonder if something special had to be done if multiple objects get predicted at the same location?

Correctness: Looks correct.

Clarity: Paper is written well.

Relation to Prior Work: Yes, paper describes relationship to past work clearly.

Reproducibility: Yes

Additional Feedback: == After Rebuttal == Thank you for providing clarifications in the rebuttal. I was already positive about the paper, and remain positive about it.


Review 3

Summary and Contributions: The paper introduces a novel approach for generating video that model the appearance and the motion of each object. Differently from existing works, the authors propose to model the object in a 3D space. A differentiable renderer is employed to generate the video from the inferred or sampled variables. The whole network is trained in a VAE fashion.

Strengths: The paper is well written. The idea is simple and produces good results on two benchmarks.

Weaknesses: The main novelty of the paper lies in the idea of including 3D in the latent space. This idea is very simple and a natural extension of existing works. One could argue that novelty is limited. There is no qualitative comparison with SOTA in the main paper. They are only in the supplementary material.

Correctness: Everything seems correct.

Clarity: Overall this is good. The related work section is very hard to read besause the names of the methods are never given. The reader needs to check the references for every citation. I would recommend to introduce the names of the methods.

Relation to Prior Work: It is clearly discussed.

Reproducibility: Yes

Additional Feedback: In Sec 2.2, I would like to know more about the motivation for using a grid rather than directly on continuous coordinates for the object. Implementing this work must be time-consuming and troublesome. The authors uploaded the code. I am convinced that releasing the code will be useful for the community and that the paper may have some impact. The code is well commented but it could be factorized better. For instance, I observed many copy-pastes between the training files for traffic and rooms. ****Post rebuttal**** After reading the other reviews and the rebuttal I maintain my rating. It seems that the complexity of the data (length, complexity of the environment, synthetic data) is a concern for some of the reviewers. To be honest, I am not familiar with the datasets employed in the paper. They look really simplistic, nevertheless, if the state of the art methods cannot work on more complicated environments, it seems hard to demand more complex ones. Therefore, I think the paper cannot be rejected for this very reason.


Review 4

Summary and Contributions: In this submission, the author(s) proposed a video generation framework by decomposing the moving scene in a video into 3D objects as well as a 3D background. A fix number of decoders are learnt to regress 3D object shapes, texture, pose information together with a decoder for regressing the background shape and texture. Then the objects and background are rendered using a differential renderer to receive 2d photometric supervision. The main contributions are: An object-centric generative model that decompose video into 3D moving objects and a 3D background. Learning to predict segmentation and tracking objects with generative model without any human annotation.

Strengths: This work capable of predict segmentation and tracking objects with generative model without any human annotation.

Weaknesses: The proposed framework consists of G fixed decoders for objects which limits the scope of the generative model to scenes with objects of a maximum number of G.

Correctness: No.

Clarity: Yes.

Relation to Prior Work: Yes.

Reproducibility: Yes

Additional Feedback: Concerning the complexity of different dataset, ROOM dataset consists of more simplex geometry with single color while TRAFFIC has relatively more realistic texture as well as complex shape like cars. However, the ROOM dataset has more diverse scene composition with more occlusion between objects while in TRAFFIC it contains basically zero occlusions between objects(at least in show cases). Further more, TRAFFIC dataset has less cars than geometry in ROOM dataset. It would be good if authors could show more results on scenes generated using CARLA with similar scene complexity level as ROOM dataset. In section 5.2, for scene generation part, the authors got quite opposite conclusions on different datasets. On ROOM dataset, GENESIS got the best FID while the O3V-mesh O3D-voxel achieved the second and third best respectively. The author(s) explained this as mesh could give shaper edges than voxel. However on TRAFFIC dataset, the mesh model yields significantly worse results comparing to voxel. It would be great if the author(s) could elaborate more on this part about this and the gap between GENESIS. ==== Post Rebuttal ==== After reading the rebuttal, I would like to change my score. The author have addressed some of my concerns. Although they are not perfectly resolved, due to the limitation of the current state-of-the-arts in this subfield. The novelty part is still not quite precise, so I am changing my overall score to 6.

[Author Response · NeurIPS 2020]

We thank the reviewers for their valuable feedback and encouraging reception of our work. We're glad they found
our 3D object-centric model of videos to be highly significant, of interest to the NeurIPS community, and potentially
impactful. We now address some points raised in the reviews; we will of course incorporate the other suggestions.

**R1: videos are short; illustrate 3D with different cameras.** We have now trained a model on 12-frame (ROOMS)
videos, showing our method can scale to significantly longer sequences. Fig. A shows generated images, objects,
and depths, which still show coherent scenes. Fig. B shows reconstructions; the rows are original, reconstructed,
reconstructed with higher camera angle (to better show the 3D structure), and objects with 3D bounding boxes (from
that angle). The additional images to the right use even more extreme viewing angles, revealing the 3D layout clearly.

**R1: low resolution & fidelity.** Training video models is expensive, particularly in 3D. We consider the resolutions
used (96x72 & 80x80) a good trade-off between computation and quality; they are also comparable with similar works
[10,11,21]. Our results show O3V-voxel produces higher-quality videos than the state-of-the-art (SCALOR) on the
complex (TRAFFIC) dataset (Tab. 2b, Fig. 5 & S6). Moreover, our method is the first that can address our tasks and
setting; it is natural that scope remains for improving visual fidelity and resolution in future work.

**R1,2,4: complexity of datasets.** Our claim of handling more visually-complex videos than prior work was meant with
reference to the state-of-the-art in generative object-centric video modelling (SCALOR) [21]; we'll clarify this. [21] is
demonstrated only on objects of near-uniform color, without shading nor perspective/3D effects. O3V successfully
models videos containing all these effects, while SCALOR fails to do so (Fig. S6 & Tab. 1). While we do only
demonstrate O3V on synthetic videos (**R2**), our (TRAFFIC) dataset contains significantly more complex videos than
state-of-the-art (**R4**)—and so we already go a significant way towards bridging the gap to natural videos.

**R4: limited number of object slots** $G$**.** Many similar models have the same limitation [21,10,29], but we agree it
would be interesting to lift it. Note $G$ is rather large in our case (tens of objects).

**R1: correctness/clarity of L86-87.** Our intended meaning is simply that the grid is 3D, with its cells placed in 3D
world-space, rather than being a 2D grid in the space of the image. Such a 2D grid is used in the generative models
[6,21], inspired by YOLO [34], which produces detections based on a grid of cells. We'll clarify in the camera-ready.

**R1: overly strong claims; inductive biases are 3D supervision.** We note that numerous related models (e.g. SQAIR,
SCALOR, IODINE), regarded by the community as unsupervised, have similar inductive biases (priors on object size,
speed, uniformity of color), albeit in 2D. We therefore respectfully disagree with this characterisation. Of course, we do
agree that inductive biases make learning easier (indeed, possible!). We will qualify the claim of compositionality at
L36 to note this refers to the generative model itself, not the encoder used to train it.

**R4: explain differing generation quality.** We'll expand the discussion in Sec. 5.2. O3V-mesh has poor FID on
(TRAFFIC) as it is prone to local optima where cars are not tracked correctly—see the images in the supplementary.
GENESIS has good FID on (ROOMS) as color segmentation (which it readily exploits) is a strong cue here.

**R1,2: use of FID.** We agree KID is a more-principled generation metric than FID; we used FID for consistency with
the works we compare to. Following **R1**'s suggestion, we re-ran our generation evaluation using KID (see Tab. A). We
see that all statements made in the paper regarding relative quality of methods remain true—in fact, FID & KID are
highly correlated on our datasets. Note (**R2**) that in calculating FID, the ground-truth feature distribution is defined by
synthetic images, so the use of a 'natural image network' is reasonable (and, indeed, common [10,28]).

**R2,3: several components of O3V (e.g. mesh renderer) are known techniques.** This is true for very many works,
and there is clearly value in showing that a novel model using some existing techniques can achieve state-of-the-art
results. Importantly, we also examine different variants (i.e. mesh vs. voxel representations), and discuss their strengths.

**R1: authors do not show scene generation results from prior works.** These are in supplementary figures S1–S6.

**R3: motivation for discrete grid of objects.** We'll expand the discussion at L90. A grid of objects ensures gradients
with respect to object locations are non-zero, even when the current predictions are poor, as at least one candidate object
should be near enough each true object. Note that the actual object locations are continuous, as they are offset by a
vector $\Delta_g$ from the cell center.

**R1: compare to GQN and SRN.** These perform novel-view synthesis, but cannot sample new scenes *a priori*. They
also cannot perform segmentation/tracking, as they lack a representation of separate objects and assume a static scene.

|  | (ROOMS) | (TRAFFIC) |
|---|---|---|
| MONet | 0.151 | 0.305 |
| GENESIS | **0.083** | 0.257 |
| SCALOR | 0.148 | 0.272 |
| O3V-voxel | 0.108 | **0.157** |
| O3V-mesh | 0.106 | 0.345 |

Table A: KID scores

Fig. A: 12-frame generations

Fig. B: 12-frame reconstructions

[Meta-Review · NeurIPS 2020]

Creating models that can perceive a rich 3D world from 2D inputs and then imagine new 3D words is both extremely difficult and important. Reviewers agreed that the model is novel, yet has many limitations. Visual fidelity is low as is resolution. The number of objects that can be rendered is small. At the same time, reviewers agreed that it outperforms prior models. Reviewers encourage the authors to move comparison with state of the art models to the main paper. The primary concern of reviewers was centered around the limitations of this sub-field as a whole, that one cannot learn from real images and generate realistic-looking images. It will take time to build models of sufficient refinement to do this, but as the reviewers point out, the work presented here provides both a conceptual advance (having a latent 3D representation) and a practical advance in generating more complex videos.